# The Effect of Psychological Distress on Measurement Invariance in Measures of Mental Wellbeing

**DOI:** 10.3390/ijerph191610072

**Published:** 2022-08-15

**Authors:** Matthew Iasiello, Eimear Muir-Cochrane, Joep van Agteren, Daniel B. Fassnacht

**Affiliations:** 1Mental Health and Wellbeing Program, Lifelong Health Theme, South Australian Health and Medical Research Institute, Adelaide 5000, Australia; 2College of Nursing and Health Science, Flinders University, Adelaide 5042, Australia; 3College of Education, Psychology, and Social Work, Flinders University, Adelaide 5042, Australia

**Keywords:** measurement invariance, wellbeing, psychological distress, assessment

## Abstract

A growing literature supports the expansion of mental health assessment to include indicators of mental wellbeing; however, the concurrent use of measures of wellbeing and distress introduces potential sources of measurement error. The current study examines whether the mental health continuum short form is invariant to the level of participants’ psychological distress. Measurement invariance testing was conducted within an Australian population (*n* = 8406) who participated in an online survey. The depression anxiety stress scale was used to construct a non-distressed group (*n* = 6420) and a severe-distress group (*n* = 1968). Results showed that metric invariance was not observed, as item loadings on the latent variables were significantly different between the groups. This signifies that wellbeing items may be interpreted and valued differently by distressed and non-distressed individuals. Metric non-invariance indicates that total and subscale scores may not be equivalent, and caution is required when making comparisons between these groups.

## 1. Introduction

A growing literature supports the need to expand the scope of traditional mental health assessment from its predominant focus on symptoms and distress to include positive states of mental health and wellbeing. By adding this focus on wellbeing, our comprehension of a person’s overall mental health is significantly improved and, subsequently, facilitates better decision-making regarding the mental health needs of the respondent. There has been academic interest in the relationship between mental wellbeing and psychological distress from as early as the 1950s when Jahoda [1] argued that the absence of disorder constituted an insufficient criterion for mental health. Empirical evidence supported this position from as early as the 1980s, with Fontana et al. [2] demonstrating that psychological health makes a unique contribution to overall mental health, independent of an individual’s degree of psychological impairment. While studies in subsequent decades continued to provide empirical support [3,4], the notion that mental wellbeing and mental illness reflect distinct continua gained significant attention following a seminal study using nationally representative data in the United States of America [5].

Keyes [5] set out to test the assumption that mental wellbeing and mental illness reflect a single bipolar dimension. Results showed that mental wellbeing and mental illness did not represent a single bipolar dimension but rather were two related yet distinct constructs. This finding has since been reproduced and replicated in more than 80 studies from around the world, using diverse assessment methods, various study methodologies, cultures, population types, and has been tested in different languages [6]. The implications of this new model of mental health stipulate the need for a concurrent focus on both mental wellbeing and mental illness and has profound implications for the way we assess mental health, promote mental wellbeing, and prevent mental illness [7,8,9].

The concurrent assessment of mental health and psychological distress introduces new considerations and challenges for anyone involved in measurement practice [10]. For example, some scales of psychological distress may already capture some aspects of mental wellbeing, and vice versa. For example, Wood et al. [11] demonstrated that a popular measure of depressive symptoms, the Centre for Epidemiologic Studies depression scale (CES-D), captures a single continuum from happiness to depression and could therefore be used to assess either construct. In the current study, we aim to investigate another consideration, which is highly relevant to the concurrent assessment of mental disorder and wellbeing, namely whether measures of mental wellbeing are invariant to participants’ level of psychological distress. Measurement invariance refers to a quality of a scale whereby “subjects from different groups with the same level on the latent variable have the same probability of obtaining equal test score” [12]. In other words, measurement invariance indicates whether a measurement captures the same construct (e.g., IQ) across different groups (e.g., males and females). For psychological research, the assumption of measurement invariance is vital as it is a prerequisite for comparing group means [13]. In the context of the dual-continua model, measurement invariance refers to the ability of measures of wellbeing to accurately assess mental wellbeing across those with or without levels of psychological distress such that scores between the two groups can be compared and meaningfully interpreted.

### 1.1. Measurement Invariance of the Mental Health Continuum Short Form

A popular assessment tool for mental wellbeing is the mental health continuum short form (MHC-SF). This measure is most commonly used in validation studies of the dual-continua model and is often used in combination with measures of psychological distress or mental illness. Previous studies have investigated the performance of the MHC-SF across demographic variables such as age, gender, and ethnicity, showing that the scale was largely invariant to these demographics e.g., [14,15,16]. To date, however, the invariance of the MHC-SF between respondents with or without psychological disorders/high distress has yet to be tested.

Measurement non-invariance occurs when different groups place different meanings on items within a scale [13,17]. Therefore, it is reasonable to hypothesize that measures of mental wellbeing may be non-invariant to psychological distress, as mental wellbeing research is inherently value-laden [18]. Further, there may be discrepancies in the ability for effective recall between those experiencing distress or not [19] or differences in post-facto evaluations of experiences. There may be differences in the ‘snap judgements’ of distressed or non-distressed participants, who are participating in the assessments rather than considered evaluations [20]. Prinzing [18] argued that “conceptions of wellbeing change dramatically even with just a little time spent in careful reflection”, which, again, could point to a difference between those experiencing psychological distress or not.

To determine measurement invariance, one moves through four levels of analysis [21]: configural invariance (i.e., related to consistent factor structures between groups), metric invariance (i.e., equality of factor loadings between groups), scalar invariance (i.e., equality of factor loadings and intercepts between groups), and strict invariance (i.e., equality of factor loadings, intercepts, and residuals) [12]. These levels of invariance are assessed sequentially, from the least constrictive, i.e., configural invariance, to the most constrictive, i.e., strict invariance.

Configural invariance is the least constrictive level of measurement invariance and tests whether a measurement tool has a similar factor structure between the two groups. If the MHC-SF does not demonstrate configural invariance, which is referred to as being non-invariant, it would indicate that the measure is tapping into different latent variables between the two groups, or that different items are loading in a different pattern on the latent variables [13]. In the case of the MHC-SF, there are three latent variables: emotional, psychological, and social wellbeing. This factor structure has been well documented in the literature, with studies demonstrating a consistent factor structure, regardless of the clinical status of the population [22,23].

If configural invariance is supported, metric invariance, the next level of measurement invariance, is assessed. This tests the degree to which individual items load on their respective latent factors. Metric non-invariance occurs when there is a significant degree of difference between the loading of the items on latent factors between the two groups. In the case of the MHC-SF, it could mean, for instance, that the item ‘During the past month, how often did you feel interested in life’ is more or less relevant to the overall ‘emotional wellbeing’ factor for clinical versus non-clinical respondents, or vice versa.

### 1.2. Aim of the Current Study

The dual-continua model provides a clear rationale for the concurrent assessment of mental wellbeing in addition to the measurement of distress and disorder. In order to do so, it is important to understand whether extant assessment tools of mental wellbeing are invariant across levels of distress such that their scores can appropriately be interpreted in both clinical and non-clinical populations. This study aims to determine measurement invariance in one such measure of wellbeing, the MHC-SF, between individuals who are highly distressed and individuals who do not display current distress. A large meta-analytic structural equation modelling study of the MHC-SF, using data from more than 100,000 participants [24], found an overall consistent-factor structure for the MHC-SF across clinical and non-clinical groups. However, there were discrepancies between item loadings for all three latent factors (i.e., emotional, psychological, and social wellbeing) of the MHC-SF in clinical and non-clinical populations, providing preliminary evidence of metric invariance within the MHC-SF. Therefore, it is hypothesized that configural invariance will be observed but that metric invariance, however, will not.

## 2. Materials and Method

### 2.1. Participants and Procedures

Participants were adults who engaged with services offered by the South Australian Health and Medical Research Institute (SAHMRI), based in Adelaide, Australia. SAHMRI is a medical health and research institute which has mental health and wellbeing as one of its focus areas. As part of its operations, it provides various wellbeing services, including internet-based measurements of mental health and wellbeing, and delivers non-specific psychological interventions to the general community.

This study relied on secondary data analysis of data collected from various SAHMRI wellbeing projects between February 2019 and April 2021. Data was collected via two recruitment streams. First, data were collected from respondents who registered for a free online mental health and wellbeing assessment via one of SAHMRI’s mental health and wellbeing websites: an online platform called the Be Well Tracker. Second, data were collected via participants who participated in SAHMRI wellbeing intervention projects, where participants could pre-register and complete the same mental health and wellbeing measurement described within the first stream prior to commencing their training. Participants for the training included individuals from the public who sought out the training via their own accord as well as people recruited for specific wellbeing projects, for example, wellbeing training provision to workforces (e.g., the private, public, or NGO sectors). The data were collected with approval by the local Human Research Ethics Committee (# 2239).

After registration, participants completed the measurement online via internet-enabled devices. It took participants approximately 10–15 min to complete the measurement, which included a range of validated mental health questionnaires, including the measures of wellbeing and distress used for this study (see ‘measures’ section below for detail). The measurement captured basic demographic information such as gender, age, employment, and study status. After completing the measurement, participants were automatically provided with their own scores and an individualized online report that explained the results and provided information about subsequent options to improve their mental health, as well as information on mental health services in case of immediate need.

A total of 8406 participants provided data for mental wellbeing and psychological distress. The mean age of the total sample was 42.1 years old (SD 13.4), with 20.4% being unemployed, while 11.4% were currently studying, as shown in Table 1. The distressed subgroup was younger than the non-distressed group and were more likely to be unemployed (χ^2^(1, 8406) = 181.66, <0.001) and/or studying (χ^2^(1, 8406) = 46.50, <0.001).

### 2.2. Measures

Mental wellbeing was measured using the MHC-SF [25]. The MHC-SF is a valid and reliable measure of mental wellbeing [24], providing a continuous measure of three key domains of wellbeing (i.e., emotional, psychological, and social well-being). The measure also facilitates an overall categorical score on whether someone has high, moderate, or low wellbeing.

Psychological distress was measured using the depression, anxiety, and stress scale-21 items (DASS-21) [26]. The DASS-21 offers reliable cut-off points for symptom severity (i.e., “mild”, “moderate”, “severe”, and “extremely severe” symptoms). Analyses were conducted using total scores for each of the three domains; internal consistencies for depression (α = 0.92), anxiety (α = 0.84), and stress (α = 0.87) were good. Participants were classified into the ‘psychological distress’ group if they scored moderate or greater distress in at least one of the three domains (i.e., Depression >7, Anxiety > 6, Stress > 10) [27].

### 2.3. Exploratory and Confirmatory Factor Analysis

All statistical analysis was conducted in SPSS v27 and AMOS v27. Exploratory factor analysis (EFA) and confirmatory factor analysis (CFA) were first conducted to explore and confirm the factor structure of the MHC-SF in the current sample. Data were randomly separated into two groups for EFA (*n* = 4233) and CFA (*n* = 4173). EFA was used on the first random sample to investigate the optimal factor structure of the MHC-SF. Parallel analysis and Scree plot inspection were used to estimate the number of factors [28]. EFA was conducted using principal axis factoring and Oblim rotation.

CFA was used on the second half of the sample to confirm the MHC-SF factor structure identified using EFA. Models tested included the original theoretical MHC-SF structure and the model identified by EFA. A good model fit was indicated by root mean square error of approximation (RMSEA) ≤ 0.05, comparative fit index (CFI) ≥ 0.97, and the highest Tucker–Lewis index (TLI) [29].

### 2.4. Invariance Testing

Invariance testing was conducted on the full sample following the protocol outlined by 32. This included beginning from the most relaxed model fit and testing invariance between the groups in increasingly strict models. CFA was used on the full sample; this time, to test that the factor structure was appropriate for non-distressed and distressed groups. The sample was split into a non-distressed (*n*= 6420) and distressed (*n*= 1968) sample, based on participants’ levels of psychological distress, who met the criteria for severe distress or greater for any of the domains on the DASS-21. Significant differences in the model fit of the distressed and non-distressed groups would violate an assumption of further invariance testing.

### 2.5. Configural Invariance Testing

Configural testing–testing whether a measurement tool has a similar factor structure between two groups–was performed using AMOS by testing the model fit *across* the two groups. The test results for a single model were tested using RMSEA, CFI, and TLI to indicate good fit (pointing to configural measurement invariance) or not (pointing to non-invariance). The model was conducted by constraining the basic factor model to one of equality across the groups. In other terms, the number of factors and their proposed indicators were held constant across the groups. All other constraints are freely estimated in each group. This model served as the baseline against which the metric invariance is compared with to test whether there is evidence of non-invariant factor loadings. In this step, the AMOS option, Emulisrel6, was selected, as suggested by [30].

### 2.6. Metric Invariance Testing

Factor loadings were constrained across groups to test whether factor loadings were invariant across the groups. Differences in factor loading scores were compared using Δχ^2^ and Δgamma hat and ΔMcDonald’s NCI. Changes in gamma hat and ΔMcDonald’s NCI are not included in AMOS; therefore, they were calculated using the calculator provided by [31]. Δgamma hat (≤0.001) and ΔMcDonald’s NCI (≤0.02) were used to indicate substantial decrements in model fit after the imposition of the equality constraints, based on the recommendations of [32].

Further steps of invariance testing did not commence, as metric invariance was not observed. Instead, possible sources of non-invariant loadings were explored by relaxing equality constraints in a stepwise way [13]. This involves testing a series of models where the equality constraint of each item is relaxed, and model fit is compared against the full metric invariance model.

## 3. Results

### 3.1. Exploratory Factor Analysis

Parallel analysis and Scree plots of the MHC-SF both suggested a three-factor model was present in the data. The pattern matrix (Table 2) was similar to the theoretically proposed solution of the MHC-SF; however, two of the social wellbeing items loaded more strongly on the psychological wellbeing factor (item 4 [social contribution] and 5 [social integration]) loaded 0.61 and 0.41 onto psychological wellbeing compared with 0.09 and 0.21 on social wellbeing, respectively. A high degree of association was observed between the three factors: emotional and social wellbeing (*r* = 0.71), emotional and psychological wellbeing (*r* = 0.85), and psychological and social wellbeing (*r* = 0.67).

### 3.2. Confirmatory Factor Analysis

CFA was used to confirm the results identified in the EFA. Models tested included the modified EFA model, as well as the originally proposed model of the MHC-SF. The original theoretical model of the MHC-SF did not fit well according to fitting indices, χ^2^ = 2785.46 (74), RMSEA = 0.094, CLI = 0.93, and TLI = 0.92, while the EFA model fit was better and within the thresholds of strong fitting, χ^2^ = 1267.14 (74), RMSEA = 0.062, CLI = 0.96, and TLI = 0.97. An adequate model fit is required to progress to the next stage; therefore, due to stronger model fitting, the modified EFA model was retained for subsequent analysis. However, please note that subsequent analyses were also conducted with the theoretically proposed model, which may be more clinically relevant, with very similar results to those reported in the main manuscript (see Appendix A).

### 3.3. Invariance Testing

The modified EFA model of the MHC-SF was tested next, with a separate CFA in the distressed vs. non-distressed groups; the results suggested that in the non-distressed sample there was a better model fit (χ^2^ = 1723.34 (74), RMSEA = 0.059, CFI = 0.97, TLI = 0.96), compared to the distressed group (χ^2^ = 709.48 (74), RMSEA = 0.066, CLI = 0.95, TLI = 0.94). While differences in χ^2^ are likely influenced by differences in sample size, RMSEA displayed substantial differences, with the distressed group showing a poorer fit (and no longer within the recommended level). However, CFA and TLI were reasonably similar and close to recommended thresholds for good fitting, allowing progression to the next stage of analysis.

### 3.4. Configural Invariance Testing

Next, the configural model was tested, which estimates the model fitting with consideration of the two groups. The results are provided in Table 3. The results indicate that the model fits the data well, RMSEA = 0.043, CFI = 0.96, TLI = 0.95, indicating that the test for configural invariance was passed; the MHC-SF factor structure was similar in both the distressed and the non-distressed groups.

### 3.5. Metric Invariance Testing

Metric invariance was subsequently assessed by constraining factor loadings to be equal across the groups. Changes in χ^2^ were significant between the metric and configural models: Δχ^2^ =108.98, df = 11, and *p* value < 0.0001. This indicated that there was a significant decrease in model fitting, suggesting that at least some factor loadings were non-invariant. This finding was supplemented by changes in gamma hat and McDonald’s NCI, which were 0.02 and 0.005, respectively, and were, therefore, within the recommended thresholds for significant change. The item loadings onto each factor across the two groups are displayed in Table 4.

To identify the items contributing to non-invariant factor loadings, the sequential analysis of model fit was tested, whereby all constraints were retained except for a single item. This process was repeated sequentially until all items were tested. Models were statistically compared to the configural model. The results are displayed in Table 5. Multiple items showed non-invariant factor loadings. These include items in the emotional wellbeing domain (items 2 [interested in life] and 3 [satisfied with life]) and the modified psychological wellbeing domain (items 4 [social contribution], 5 [social integration], 11 [positive relations with others], and 12 [personal growth]). Invariance testing was concluded at this point due to the high proportion of non-invariant factor loadings, particularly as two-thirds of the items of the emotional wellbeing latent factor and 50% of the items on the modified psychological wellbeing factor were non-invariant.

## 4. Discussion

The current study assessed the measurement invariance of the MHC-SF for distressed versus non-distressed individuals. The results suggest that while the tripartite factor structure of the MHC-SF is consistent across those who had moderate–severe psychological distress (configural invariance), the degree to which the items were loaded onto the latent factors was different between those with low versus high levels of distress (metric non-invariance).

While EFA and CFA supported the originally proposed three-factor model of the MHC-SF [25], two items (assessing social contribution and social integration) from the original social wellbeing scale loaded more strongly onto the psychological wellbeing scale. There is apparent face validity to this result, as the two items could be understood as psychological functioning within the community (e.g., item 4, “I have something to contribute to society”), whereas the remaining three social wellbeing items were more akin to the objective evaluation of society in general and whether it is ‘just’ to its members (e.g., “Society is a good place”). As the majority of studies on the relationship between mental illness and mental health assume the theorized factor structures of scales used, e.g., those within [33], this is a precautionary reminder that model fitting can change between samples; even in a tool as widely validated as the MHC-SF.

The modified factor structure was supported by CFA in both the distressed and non-distressed groups. While the fit was poorer in the distressed group, it was still within the acceptable limits. This finding indicates that the MHC-SF displayed configural invariance across the two groups, meaning that the same factor structure was retained across both samples. This is in line with prior research demonstrating that the factor structure of the scale is similar across both clinical and non-clinical groups [22,23,24]. Despite the finding of metric non-invariance (discussed below), responses from distressed and non-distressed individuals indicated the presence of emotional wellbeing, psychological wellbeing, and social wellbeing.

This study is the first, to our knowledge, to go a step further from configural invariance and investigate metric invariance between the two groups. The findings showed that several items contributed to metric non-invariance, meaning that their degree of loading onto their respective factors was different between the two groups. Importantly, this does not suggest that the actual responses differed between distressed versus non-distressed groups (i.e., being higher or lower between groups) but that the pattern of participants’ scores was different across the two groups (i.e., being interested in life is more relevant for emotional wellbeing in non-distressed individuals compared with distressed ones). This is an important finding, as previous studies concluded that measures are appropriate in both clinical and non-clinical groups based on the factor structure alone, while this study demonstrates that a measure may be non-invariant, even when the factor structure fits in both groups.

Measurement invariance is not a black and white issue, and it is possible for scales to be considered ‘partially invariant’. Standards of partial invariance vary in the literature, with recommendations suggesting that the use of scales remains appropriate when, ideally, more than half of the items loading onto a factor should be invariant [13,34]. In this study, metric invariance was found within a large number of items, affecting the majority of items in the emotional wellbeing and psychological wellbeing factors. For this reason, we conclude that, in this sample, and with the modified factor structure, the MHC-SF should not be considered partially invariant.

Our analyses worked with a modified factor structure, as the EFA indicated a superior fit compared to the original factor structure by letting two items load onto a different factor. Metric invariance was also tested in this sample using the original factor structure of the MHC-SF (see Appendix A). While the results for the analyses using the original model suggested that partial invariance may be acceptable, as a minority of items per factor were non-invariant, this analysis should be conducted within populations where the original model represents the most appropriate fit [30].

The results of this study have important implications for interpretation of the scores of the MHC-SF. Chen [35] showed that bias in the latent variable means increases as the percentage of non-variant items loading onto a latent factor increases; therefore, caution should be considered when interpreting and comparing latent mean scores between the groups with and without psychological distress. Metric invariance is an issue for comparing latent variable scores between different groups but also affects the accuracy of categorical and predictive scoring [12]. For example, the impact of metric invariance has been assessed using Monte Carlo simulation, suggesting that metric invariance has significant negative effects on the predictive validity of scales [12]. Results showed that the reliability of the scale decreased due to non-discriminating items (metric non-invariant items), which in turn affected the likelihood that cut-off points and diagnoses become inaccurate. This may likely affect the categorical accuracy of the MHC-SF when comparing groups of distressed and non-distressed participants, as the emotional wellbeing items (which were mostly non-invariant) are very influential in the categorization scoring process.

Metric non-invariance, as observed in the current study, signifies that there is a difference in the loading of items onto their latent factor between two different groups. This can be due to different values being placed on the items between groups. Metric non-invariance in the current study might be related to the value-laden nature of the wellbeing items and that these items were answered differentially across the groups. For the emotional wellbeing latent variable, the largest source of metric invariance was the second item, which asked participants how often they felt “interested in life”; this suggests that for emotional wellbeing participants (with vs. without high levels of distress) responded to this item most differentially. Specifically, the item of being ‘interested in life’ was more strongly relevant to emotional wellbeing for those with high psychological distress than those without. On the modified psychological scale, causes of metric invariance mainly came from items related to personal growth (“that you had experiences that challenged you to grow and become a better person”), relationships with others (“that you had warm and trusting relationships with others”), and from the two items from the original factor model social wellbeing variable about contribution (“that you had something important to contribute to society”), and belonging (“you belonged to a community, like a social group, or your neighborhood)”.

The current study was not designed to investigate the reasons for metric invariance in these items, although various potential explanations can be given. First, it has been commented that wellbeing items are inherently value-laden [18], and therefore it stands to reason that those experiencing psychological distress place a differential weight on certain items than those not experiencing distress. Growth, relationships with others, contribution, and belonging are certainly value-laden and central to concepts such as psychological safety [36], psychological needs [37], and self-determination theory [38]; however, this potential cause does not sufficiently explain why some items were found to be non-invariant and not others. The relevance of mental wellbeing has been investigated in individuals experiencing mental illness [39,40], and future research could apply these findings to improving scale performance in the future.

Second, the metric invariance of these items may be influenced by affective recall bias, as it has been established that people are inaccurate in their recollection of past affective experiences in the context of distress [41]. Colombo et al. [42] used ecological momentary analysis to demonstrate that those with mild depressive symptoms tended to overestimate negative affective experiences, while those without distress overestimated positive affective experiences. It is possible that this bias impacted metric invariance in the current study, as it is possible that many of these items have a positive valence which could be under or overestimated, depending on current levels of psychological distress.

Third, it has been demonstrated that non-effortful reporting is associated with errors in measurement invariance testing [43]. Non-effortful reporting, as measured by the time taken to complete a survey tool, can lead to biased factor loading estimates that directly impact metric invariance [43]. There may be an issue in how much time is spent considering the items of a wellbeing scale between those experiencing distress and those who are not. It is reasonable to consider that some items related to concepts as psychologically important as growth, relationships, and belonging may be sensitive topics for someone experiencing psychological distress, which leads to less time being spent considering them.

### 4.1. Implications for Theory

The current study has important implications for wellbeing and measurement theory. First, there is a need to consider the impact of psychological distress on the validity of latent variable scores in wellbeing measures. Differences in the pattern of responses, due to metric-invariance, can influence the latent scores across the two groups. As a result, hesitation and caution may be required when comparing latent variable scores from groups with different psychological distress or mental illness profiles. For example, researchers may notice differences in the wellbeing of two groups which are artifacts of the level of distress in the two groups. While the results from this study support those identified at the meta-analytic level for the MHC-SF [24], more research is needed to confirm if measurement invariance is an issue in other populations using different measures of wellbeing and distress. Further research is required to understand the primary source(s) of metric invariance between clinical or distressed and non-distressed groups. As discussed above, these sources may derive from the value-laden nature of wellbeing surveys, affective recall bias, or non-effortful reporting.

The findings from this study support previous research, confirming the factor structure of MHC-SF into emotional, psychological, and social factors, regardless of psychological distress [24]. This suggests that the assessment of mental wellbeing is relevant despite the presence of distress, and scales such as the MHC-SF could be used to assess changes in these aspects of wellbeing in clinical settings. Finally, the results from the EFA and CFA present a reminder that the factor structure of even well-validated scales may not be present in particular samples and should be tested rather than assumed.

### 4.2. Limitations and Future Directions

This study was limited by the sample and assessment tools utilized. The study was conducted in the general community using a general measure of psychological distress; therefore, the results of the ‘distressed group’ cannot necessarily be generalized to clinical populations. Further, the use of the DASS cut-points to divide participants into those with low vs. high levels of distress is arbitrary; thus, future studies need to replicate these findings in populations that are known to be highly distressed. While the cut-points lead the differently sized samples, previous research has demonstrated that sample sizes over 400 show uniformly high precision of estimated-factor-loading differences; therefore, differences in sample sizes are not anticipated to cause an issue [44]. It is possible that non-invariance becomes stronger with greater distress or in clinical populations, as observed by Iasiello et al. [24]. The study was not designed to investigate the causes of the observed non-invariance, and features of the sample may have acted as confounding factors. For example, the online, self-selecting recruitment method may disproportionately identify participants who are more interested or motivated to focus on their mental health, excluding older participants and explaining the demographic differences between the two groups (in particular, the fact that distressed groups were more likely to be students). Future studies could also endeavor to modify problematic items that lead to measurement non-invariance in distressed samples by potentially clarifying or reducing the degree to which they are ‘value-laden’. Alternatively, different analysis techniques could be used on appropriate datasets, such as multi-level CFA modelling or multi-group measurement invariance analysis. Future studies should investigate the potential causes of metric invariance in these groups such that wellbeing measures can be modified and improved to avoid this source of measurement error.

## 5. Conclusions

The current study aimed to test the measurement invariance of the MHC-SF between individuals experiencing high levels of psychological distress and non-distressed individuals. In both groups, it was found that the MHC-SF taps into three domains of mental wellbeing: emotional, psychological, and social wellbeing. However, it was identified that there were differences in the item loadings on each of these latent variables between the two groups. This signifies that there may be differences in the way that these items are valued or interpreted, and that caution is needed when comparing wellbeing scores between groups who are experiencing psychological distress or not.

## Figures and Tables

**Table 1 ijerph-19-10072-t001:** Summary demographics of participants, by distressed and non-distressed group.

	Total (*n* = 8406)	Non-Distressed (*n* = 6420)	Distressed (*n* = 1986)
Age (years), mean (SD)	42.1 (13.4)	43.5 (13.4)	37.7 (13.4)
Gender (female), *n* (%)	4104 (48.8)	3183 (49.6)	920 (46.3)
Unemployed, *n* (%)	1716 (20.4)	1099 (17.1)	617 (31.1)
Studying, *n* (%)	959 (11.4)	649 (10.1)	311 (15.7)

**Table 2 ijerph-19-10072-t002:** Exploratory factor analysis pattern matrix results in random samples of participants (*n* = 4233).

Item	Emotional Wellbeing	Social Wellbeing	Psychological Wellbeing
1 happy	0.82		
2 interested in life	0.73		
3 satisfied with life	0.74		
4 that you had something important to contribute to society			0.56
5 that you belonged to a community (like a social group, or your neighborhood)			0.32
6 that our society is a good place, or is becoming a better place, for all people		0.82	
7 that people are basically good		0.74	
8 that the way our society works makes sense to you		0.80	
9 that you liked most parts of your personality			0.72
10 good at managing the responsibilities of your daily life			0.66
11 that you had warm and trusting relationships with others			0.61
12 that you had experiences that challenged you to grow and become a better person			0.73
13 confident to think or express your own ideas and opinions			0.81
14 that your life has a sense of direction or meaning to it			0.66

**Table 3 ijerph-19-10072-t003:** Results of measurement invariance testing in the modified MHC-SF factor structure.

Model	χ^2^	RMSEA	CFI	TLI	Δχ^2^
Configural	2138.50 (148)	0.043	0.96	0.95	59.41 *p* < 0.001
Metric	2197.91 (159)	0.042	0.96	0.96	

**Table 4 ijerph-19-10072-t004:** Item loadings onto the modified EFA MHC-SF factor structure for the highly distressed and non-distressed groups.

Item	Factor	Item Loading	
		Non-Distressed	Distressed
1	Emotional wellbeing	0.75	0.73
2	Emotional wellbeing	0.80	0.83
3	Emotional wellbeing	0.83	0.83
4	Psychological wellbeing	0.84	0.85
5	Psychological wellbeing	0.78	0.69
6	Social wellbeing	0.79	0.75
7	Social wellbeing	0.73	0.71
8	Social wellbeing	0.66	0.64
9	Psychological wellbeing	0.73	0.68
10	Psychological wellbeing	0.61	0.54
11	Psychological wellbeing	0.69	0.65
12	Psychological wellbeing	0.63	0.69
13	Psychological wellbeing	0.68	0.63
14	Psychological wellbeing	0.82	0.81

**Table 5 ijerph-19-10072-t005:** Identification of the source of metric invariance in the modified MHC-SF factor structure.

Relaxed Item:	χ^2^	df	Δχ^2^	Δdf	*p*
1	2196.24	158	1.66	1	0.198
2	2179.20	158	18.70	1	<0.001
3	2189.50	158	8.40	1	0.004
4	2193.98	158	3.92	1	0.048
5	2182.95	158	14.95	1	<0.001
6	2196.90	158	1.00	1	0.317
7	2197.80	158	0.10	1	0.752
8	2197.90	158	0.00	1	1.000
9	2196.94	158	0.96	1	0.327
10	2197.90	158	0.00	1	1.000
11	2192.85	158	5.05	1	0.025
12	2180.50	158	17.40	1	<0.001
13	2197.40	158	0.50	1	0.479
14	2197.50	158	0.40	1	0.527

## Data Availability

Data available upon reasonable request.

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
