# Peer review of "The Effect of Psychological Distress on Measurement Invariance in Measures of Mental Wellbeing"

_ijerph, 2022, doi:10.3390/ijerph191610072_

Round 1
Reviewer 1 Report
I had the pleasure to read the paper entitled "The effect of psychological distress on measurement invariance in measures of mental wellbeing", which implemented Confirmatory Factor Analysis (CFA) and the Invariance Testing, in order to investigate the differences between the distressed vs. non-distressed groups. In addition, the paper shows that wellbeing items may be interpreted and valued differently by distressed and non-distressed individuals. But I also had several concerns about this manuscript.
1. Line 24-61, this part of the content is weakly related to the aim of the article and can be appropriately shortened. This section mainly cites literature to illustrate the range of mental health assessments from focusing primarily on symptoms and distress to a positive state including well-being. But the article aims to explore whether the measurement of mental health differs across the distressed vs. non-distressed groups. Many cited articles here may be deleted.
2. Line 85-94, this section did not give sufficient information and discussions related to the research topic. Therefore, the content of this section should be appropriately expanded.
3. Lines 137-154, This article applied two sources of data: users registered with the South Australian Institute of Health and Medical Research (SAHMRI) and participants in the (SAHMRI) health intervention program. The collection of data is not random. The users registered in this institute are mainly researchers, people with physical and mental illness, people who are more concerned about their own health and other specific groups. Thus, the sampling problem should be discussed in more detail.
4. Online survey may not include sufficient old people who are excluded on the Internet.
5. The multi-level CFA model may be also good for testing the difference across the distressed vs. non-distressed groups.
6. Lines 311-322, the recitation in the conclusion is too long. It is recommended to use a more concise language here.
Author Response
Thank you for reviewing out paper, we have addressed all of your comments and feel that that manuscript is now stronger for it. Please see our responses below:
1. Large segments of this section have been condensed. We argue there is a need to introduce the evidence of the dual-continua model to justify the assessment of wellbeing in the context of distress/mental illness. This is because there remains a large cohort of researchers in the field who still consider wellbeing and illness opposites (therefore only needed to measure one dimension or the other).
2. This section has been expanded and re-structured to provide more relevant information for the research topic.
3. This sampling issue has been raised in the limitations section of the paper
4. This sampling issue was also described in the limitations
5. This suggestion has been included in future directions
6. The introduction of the conclusion have been reduced in line with your comment.
A thorough spell check was also conducted.

Reviewer 2 Report
I really enjoyed reading this valuable manuscript. I believe the results of the statistical invariance model may provide meaningful implications regarding the expansion of mental health assessment in clinical contexts. However, I found several minor and major points that need to be reconsidered in the current format.
I understand there is a paucity of research highlighting the proposed invariance model. However, the clear rationale why testing the modified invariance model is significant in the given (clinical) context should be discussed in the Introduction section
In the Literature section, I believe the research hypothesis is interesting and makes sense. For better quality of this manuscript, a concrete discussion that helps develop the hypothesis (configural invariance will be observed, however, that 133 metric invariance will not) in the given (clinical) setting.
Please check if there could be any statistical biases about the large gap of demographics ratio in distressed (about 73%, n= 6,420) and non-distressed group (about 27%, 1,986).
In the Method section, I think the authors can integrate both “2.2. Analysis” and “2.3. Exploratory and confirmatory factor analysis” in a section.
Personally, attaching the measurement items (e.g., the MHC-SF) used may enable for better understanding the latent variables in the proposed model.
I am fine with the Implication section – however, I expected unique insights of the implication based on the results.
Thank you!
Author Response
Thank you for reviewing out paper, we have addressed all of your comments and feel that that manuscript is now stronger for it. Please see our responses below:
1. A rationale for the modified factor structure has been added to the Methods section. We agree that the original structure may be more clinically relevant, and have added more clarity as to why it was included in the supplementary materials.
2. The introduction has been restructured to provide a stronger justification of the hypothesis much closer to the relevant section, providing greater clarity for the formulation of the hypothesis.
3. This potential issue has bolstered with an added reference and described in the limitation section.
4. These sections have been merged
5. The measurement items have been added to Table 2
6. A concrete example has been added to this section to provide detail of the more ‘unique’ insights into the implications of this work.
The manuscript has also benefited from a close read through for style and language check.
